## Research Article

mental health; barriers; developing countries; global mental health; psychiatrist

**Corresponding author:**
Rodrigo Ramalho;
Email: r.ramalho@auckland.ac.nz

# Mental health research in South America: Psychiatrists and psychiatry trainees' perceived resources and barriers

Rodrigo Ramalho[1] , Vanessa Chappe[2], Lisette Alvarez[3] ,
Gianfranco C.A. Argomedo-Ramos[4] , Guillermo Rivera Arroyo[5] ,
Graciela L. Bonay[6] , Javiera C. Libuy Mena[7] , Miguel A. Cuellar Hoppe[8] ,
Domenica N. Cevallos-Robalino[9] and Jairo M. Gonzalez-Diaz[10]

[1]Department of Social and Community Health, School of Population Health, The University of Auckland, Auckland, New Zealand; [2]Administración de los Servicios de Salud del Estado (ASSE), Montevideo, Uruguay; [3]Hospital Regional Daniel Alcides Carrión, Cerro de Pasco, Perú; [4]Facultad de Medicina de la Universidad Nacional de Trujillo, Trujillo, Perú; [5]Deparment of Psychology, Universidad Privada de Santa Cruz de la Sierra, Santa Cruz, Bolivia; [6]Private Practice, Rosario, Argentina; [7]Department of Psychiatry, School of Medicine, Pontificia Universidad Católica de Chile, Santiago, Chile; [8]Centro integral de Neurociencias, Asunción – Paraguay, Asunción, Paraguay; [9]Epidemiology and Public Health, UASB, Quito, Ecuador and [10]UR Center for Mental Health – CeRSaMe, School of Medicine and Health Sciences, Universidad del Rosario, Bogotá, Colombia

## Abstract

As mental health issues continue to rise in Latin America, the need for research in this field becomes increasingly pressing. This study aimed to explore the perceived barriers and resources for research and publications among psychiatrists and psychiatry trainees from nine Spanish-speaking countries in South America. Data was collected through an anonymous online survey and analyzed using descriptive methods and the SPSS Statistical package. In total, 214 responses were analyzed. Among the participating psychiatrists, 61.8% reported having led a research project and 74.7% of them reported having led an academic publication. As for the psychiatry trainees, 26% reported having conducted research and 41.5% reported having published or attempted to publish an academic paper. When available, having access to research training, protected research time and mentorship opportunities were significant resources for research. Further support is needed in terms of funding, training, protected research time and mentorship opportunities. However, despite their efforts to participate in the global mental health discussion, Latin American psychiatrists and psychiatry trainees remain largely underrepresented in the literature.

## Impact statement

Even though mental health issues continue to rise in Latin America, the region remains constantly underrepresented in the global mental health discussion. In this article, we discuss the barriers to and resources for research and research dissemination, as reported by psychiatrists and psychiatry trainees from nine Spanish-speaking South American countries. Our findings highlight how access to research training, mentorship opportunities and protected research time are significant resources when available. At the same time, our results show how lack of funding, protected research time, research training and mentorship opportunities are prevalent barriers in the region, all of which have a significant impact on research endeavors. Despite the lack of resources, Latin American psychiatrists and psychiatry trainees are keen to participate in the global mental health discussion and appear to be taking active steps toward doing so. However, further support is needed to ensure the production, dissemination and translation of research that will help meet the region's mental health needs.

## Introduction

Mental health is a rapidly evolving field that relies on research and research dissemination to inform care and policy needs, as well as to develop innovative and cost-effective individual and public health responses. However, conducting research in mental health is often not without its challenges, particularly in low- and middle-income countries, where resources for research are often scarce.

In Latin America, the growing burden of mental health disorders (Kohn et al., 2005; Caldas de Almeida and Horvitz-Lennon, 2010) is mirrored by an equally growing and urgent need for

mental health-related research in the region (Alarcon, 2003). Latin America has remained largely underrepresented in the global mental health-related literature. In 2010, Gallo and Tohen (2010) reported that Latin America contributed only 2% of the world's mental health literature. During the COVID-19 pandemic, Akintunde et al. (2021) found a growing contribution to this literature from numerous countries around the world, including Latin American countries. However, Yalcin et al. (2022) reported that only about 4% of this literature was led by authors from Latin America and the Caribbean.

Previous authors have highlighted the need for further mental health-related research in Latin America (Razzouk et al., 2008; Kohn et al., 2018). Addressing existing gaps is essential to promote mental health and well-being in the region. However, it is important first to understand what has hindered mental health research in Latin America and identify potential strategies to overcome these obstacles. Responding to this need, the present study aimed to explore the barriers and resources for research and publications among psychiatrists and psychiatry trainees in nine South American countries.

## Methods

This was a cross-sectional descriptive observational study conducted in August 2021. The research team comprised psychiatrists and psychiatry trainees from nine countries in South America, namely Argentina, Bolivia, Chile, Colombia, Ecuador, Paraguay, Perú, Uruguay and Venezuela. Informed by a literature review, the research team developed an online anonymous survey using Qualtrics. The survey was developed in Spanish and comprised three main sections: demographic data, barriers and resources for research and barriers to publication. The sections on barriers and resources for research and barriers to publication presented participants with a list of options and asked them to select the one/s that applied to them. It also offered participants the possibility to add to the list via free text.

An invitation to participate in the study with a hyperlink to the survey was distributed among the research team's professional networks via email and social media. Psychiatrists and psychiatry trainees working or studying in Spanish-speaking South American countries, that is, Argentina, Bolivia, Chile, Colombia, Ecuador, Paraguay, Peru, Uruguay and Venezuela, were invited to participate. No exclusion criteria were used. Ethical approval was obtained from the Clínica Nuestra Señora de la Paz's Ethics Committee, Bogota, Colombia, on July 14, 2021. The first page of the survey asked participants to consent to the study before accessing the survey and provided additional information about the study, including the first author's contact information. Data collection occurred between August 1, 2021, and September 1, 2021. The collected data was stored in the Qualtrics software, which provided an initial descriptive analysis. The data was further analyzed using the SPSS v.25 program for Windows. The relationships between categorical variables were tested using bivariate analysis with a chi-square test, with a *p*-value < 0.05 accepted as significant. Several logistic regression models were performed independently for psychiatrists and psychiatry trainees. In these models, participation in scientific publications (for both groups) and scientific research in general (for psychiatrists) were explored as independent variables controlled by explanatory covariates that had shown a statistically significant association in the bivariate analyses. Even though gender did not show a statistically significant relationship in the

bivariate analysis, we included it in the logistic regression models given its known impact on science representation.

## Results

A total of 214 responses were analyzed. Table 1 presents the demographic characteristics of the total sample. Most participating psychiatrists were female, between 31 and 40 years old, and held their specialization as their highest academic degree. The countries with the most representation of psychiatrists were Uruguay, Venezuela and Bolivia, comprising about three-quarters of the total sample of psychiatrists. Most psychiatrists ($n = 47$; 27.2%) reported having 1–5 years of working experience as psychiatrists (vs. $n = 41$; 23.7% with >25 years; $n = 30$; 17.3% with 6–10 years; $n = 20$; 11.6% with 11–15 years; $n = 18$; 10.4% with 16–20 years and $n = 17$; 9.8% with 21–25 years of working experience).

Similarly, most participating psychiatry trainees were also female, between 31 and 40 years old, and most of them held medical doctor degrees as their highest academic degree. The countries with the most representation of psychiatry trainees were Uruguay, Peru and Argentina, comprising about half of the total sample. Most psychiatry trainees reported studying at a public university ($n = 33$; 80.5% vs. $n = 8$; 19.5% in private universities). Overall, four of the nine participating countries – Uruguay, Venezuela, Peru and Bolivia – contributed with slightly over three-quarters of the total sample of psychiatrists and psychiatry trainees. Similarly, about 66% of the total sample self-identified as female.

When asked whether conducting research in psychiatry is important, most psychiatrists ($n = 165$; 95.4%) and psychiatry trainees ($n = 38$; 92.7%) responded affirmatively. More than half of the total samples of psychiatrists and psychiatry trainees reported having received research training, mainly at a postgraduate level (see Table 2). More than half of the total sample of psychiatrists (61.8%) reported having conducted a research project, versus 26.8% of psychiatry trainees. Additionally, 59.8% of those psychiatrists who conducted research reported having led it, versus 12.2% of psychiatry trainees. About three-quarters of the psychiatrists who reported having led a research project did so without financial support. Practically half of the total sample of psychiatrists reported having published an article in an academic journal, mainly as a first, last or corresponding author. On the other hand, more than half of the total sample of psychiatry trainees reported not having published an article in an academic journal.

Table 3 shows barriers to and resources for research and barriers to publishing. When exploring available resources for research, we found that a third of the sample reported time as an available resource, while a minority of participants reported financial support. The main barriers to research were lack of incentives or recognition, lack of financial support and lack of time. The main barriers to publishing were lack of time, financial support and mentorship.

Among the psychiatrists, research training was significantly higher in participants who had conducted a research project ($\chi^2 = 4.554$; $p = 0.024$) and in those who had led a research project ($\chi^2 = 12.680$; $p < 0.001$). Regarding available resources for research, a statistically significant relationship was found between having conducted research and having had time to conduct research ($\chi^2 = 3.853$; $p = 0.035$) and mentorship ($\chi^2 = 4.532$; $p = 0.024$). Likewise, several barriers to research showed a statistically significant relationship with this same outcome, namely: lack of time to research or publish ($\chi^2 = 5.212$; $p = 0.016$;

**Table 1.** Demographic characteristics

| Category | Variables | Psychiatrists | | Trainees | |
|---|---|---|---|---|---|
| | | *n* | % | *n* | % |
| Gender | Female | 108 | 62.4 | 33 | 80.5 |
| | Male | 65 | 37.6 | 8 | 19.5 |
| | Nonbinary | 0.0 | 0.0 | 0.0 | 0.0 |
| Age | 20–30 | 4 | 2.3 | 16 | 39.0 |
| | 31–40 | 57 | 32.9 | 21 | 51.2 |
| | 41–50 | 38 | 22.0 | 2 | 4.9 |
| | 51–60 | 40 | 23.1 | 2 | 4.9 |
| | >60 | 34 | 19.7 | 0.0 | 0.0 |
| Country of residence | Argentina | 14 | 8.1 | 6 | 14.6 |
| | Bolivia | 25 | 14.5 | 5 | 12.2 |
| | Chile | 12 | 6.9 | 1 | 2.4 |
| | Colombia | 10 | 5.8 | 1 | 2.4 |
| | Ecuador | 2 | 1.2 | 0.0 | 0.0 |
| | Paraguay | 2 | 1.2 | 0.0 | 0.0 |
| | Perú | 28 | 16.2 | 8 | 19.5 |
| | Uruguay | 41 | 23.7 | 16 | 39.0 |
| | Venezuela | 39 | 22.5 | 4 | 9.8 |
| Highest academic degree | Medical doctor | 0.0 | 0.0 | 31 | 75.6 |
| | Specialization | 133 | 76.9 | 8 | 19.5 |
| | Master | 28 | 16.2 | 1 | 2.4 |
| | PhD | 12 | 6.9 | 1 | 2.4 |

$\chi^2 = 10.151$; $p = 0.001$) and lack of financial resources for research or publishing ($\chi^2 = 4.302$; $p = 0.028$; $\chi^2 = 3.331$; $p = 0.048$). Those who had conducted research projects had also participated in scientific publications ($\chi^2 = 25.092$; $p < 0.001$). On the other hand, lack of interest ($\chi^2 = 5.581$; $p = 0.019$), difficulties in choosing a scientific journal ($\chi^2 = 3.566$; $p = 0.042$), lack of mentoring for scientific publications ($\chi^2 = 3.325$; $p = 0.048$) and lack of funding to publish ($\chi^2 = 7.385$; $p = 0.005$) constituted the main barriers to participate in scientific publications.

Two logistic regression models were carried out (Table 4), including all the variables that showed a statistically significant relationship with the dependent variables "Having conducted research" and "Scientific publications" (which included having published and having attempted to publish). In the first case, it was found that lack of financial resources was a barrier to conducting research ($p = 0.013$, ORadjusted = 2.080, 95% CI 1.167–3.706). On the other hand, having conducted research was the main factor related to having published or attempted to publish ($p = 0.033$, ORadjusted = 0.431, 95% CI 0.198–0.935). In neither of the two cases, gender was found to have an impact on the findings ($p > 0.05$, ORadjusted = 0.863, 95% CI 0.435–1.713 and $p > 0.05$, ORadjusted = 0.723, 95% CI 0.344–1.519).

Among psychiatry trainees, a statistically significant relationship was only found between having published or attempted to publish and two barriers to publication: limited knowledge of English ($\chi^2 = 4.490$; $p = 0.035$) and lack of funding ($\chi^2 = 6.470$; $p = 0.012$). A logistic regression model was performed (Table 4),

including the two variables that showed a statistically significant relationship with the dependent variable "Scientific publications" (which included having published or attempted to publish). It was found that the lack of funding for publication was the main barrier to scientific publications ($p = 0.025$, ORadjusted = 0.076, 95% CI 0.008–0.722). In this case, gender also did not show a statistically significant relationship with the dependent variable of interest ($p > 0.05$, ORadjusted = 0.158, 95% CI 0.014–1.776).

## Discussion

As mental health issues continue to rise in Latin America, the need for research and innovation in the region becomes more pressing. The present study explored barriers and resources for research and research dissemination among psychiatrists and psychiatry trainees in Spanish-speaking countries in South America. Most participating psychiatrists reported having conducted research, with more than half reporting having led a research project. Similarly, most participating psychiatrists reported having published in academic journals, with about three-quarters reporting having led a publication. Having carried out research projects was significantly associated with having received research training, having time to research, and having mentorships as available resources. Additionally, participating in research was the main factor related to having published or attempted to publish. Lack of funding was one of the main barriers to research and publications.

**Table 2.** Research and publishing experience

| Category | Variables | Psychiatrists | | Trainees | |
|---|---|---|---|---|---|
| | | *n* | % | *n* | % |
| Is it important to conduct research in psychiatry? | Yes | 165 | 95.4 | 38 | 92.7 |
| | No | 8 | 4.6 | 3 | 7.3 |
| Research training | Yes | 104 | 60.1 | 28 | 68.3 |
| | No | 69 | 39.9 | 13 | 31.7 |
| If yes, when did you receive that training? | Undergraduate | 45 | 43.3 | 13 | 31.7 |
| | Postgraduate | 52 | 50.0 | 14 | 34.1 |
| | Independent | 7 | 6.7 | 1 | 2.4 |
| Have you conducted a research project? | Yes | 107 | 61.8 | 11 | 26.8 |
| | No | 66 | 38.2 | 30 | 73.2 |
| If yes, have you led that research project? | Yes | 64 | 59.8 | 5 | 12.2 |
| | No | 44 | 40.2 | 6 | 14.6 |
| Was the project that you led funded? | Yes | 15 | 23.4 | 2 | 4.9 |
| | No | 49 | 76.6 | 3 | 7.3 |
| Have you published in an academic journal? | Yes | 91 | 52.6 | 15 | 36.6 |
| | I attempted to | 20 | 11.6 | 2 | 4.9 |
| | No | 62 | 35.8 | 24 | 58.5 |
| If yes, have you published as a first, last or corresponding author? | Yes | 68 | 74.7 | 4 | 9.8 |
| | No | 23 | 25.3 | 11 | 26.8 |

Latin American psychiatrists seem to be conducting and leading research and academic publications. However, authors like Gallo and Tohen (2010) and Yalcin et al. (2022) found that Latin American researchers remain largely underrepresented in the global mental health literature. It is relevant to notice that both Gallo and Tohen and Yalcin and colleagues' studies focused on publications made in the English language. So, while Latin American psychiatrists appear to be contributing to the mental health literature, they are likely relying on international collaborations when publishing in English, although not as leading authors (Grácio et al., 2020).

Participants reported various barriers to conducting research and publishing in academic journals. Lack of financial support, incentives or recognition and protected research time were the main barriers to research among participating psychiatrists and psychiatry trainees. Similarly, lack of financial support, time and mentorship – particularly for psychiatry trainees – were reported as the main barriers to publishing in academic journals. Limited English proficiency was also a barrier to publishing. As previous authors have suggested, language may act as a multilevel barrier in mental health research (Ransing et al., 2023). That is, non-English speaking researchers may face barriers not only at a dissemination level but also in terms of accessing research resources, such as screening tools and assessment batteries.

Lack of funding for mental health is a significant challenge in many Latin American countries (Rodríguez, 2010; Caldas de Almeida, 2013), and health research has been traditionally underfunded in the region (Moloney, 2009). Many countries in the region often have limited budgets for healthcare, and mental health is, in many cases, even further underfunded. This very often translates into psychiatrists having limited resources to conduct research,

including time as a resource. Plus, the urgency and severity of regional or national mental health needs often demand psychiatrists to prioritize providing clinical care over research.

Only about a quarter of psychiatry trainees reported having conducted research, and less than half reported having published or attempted to publish. Our findings support the call to seek ways to increase access to training and mentorship opportunities for psychiatry trainees and early career psychiatrists (El Halabi et al., 2021; Naskar et al., 2022). Mentoring relationships could prove doubly beneficial, providing both mentors and mentees with the possibility to learn and grow from each other's strengths, talents and experiences (Ramalho et al., 2016). Online training or mentorships could provide access to these resources when unavailable at a local or regional level (Naskar et al., 2022). However, if offered from outside of Latin America, participants based in Latin America should lead the focus of these programs. Mentees in Latin America should not become a resource to be utilized for the benefit of non-Latin America-focused research agendas. Peer-led learning models could also provide psychiatry trainees and early career psychiatrists opportunities to collaborate in training, research projects and research dissemination (Ransing et al., 2021).

While gender did not emerge as a significant factor in the present study, it remains a critical dimension to be explored in future research endeavors. Even though Latin America is a vast and diverse region, it has a distinct cultural, social and economic context where gender dynamics may play an impactful role in research participation, career trajectories and publication opportunities for psychiatrists and psychiatry trainees of different genders. Previous authors have found that despite notable progress, there is still a lack of adequate representation of women in the field of academic

**Table 3.** Resources for and barriers to research and publication

| Category | Variables | Psychiatrists | | Trainees | |
|---|---|---|---|---|---|
| | | n | % | n | % |
| Resources for research | Time | 63 | 36.4 | 8 | 19.5 |
| | Training | 32 | 18.5 | 9 | 22.0 |
| | Technical support | 25 | 14.5 | 5 | 12.2 |
| | Mentorship | 56 | 32.4 | 22 | 53.7 |
| | Others | 20 | 11.6 | 1 | 2.4 |
| | Financial support | 12 | 6.9 | | |
| Barriers to research | Lack of incentives or recognition | 101 | 58.4 | 31 | 75.6 |
| | Lack of financial support | 102 | 58.3 | 25 | 61.0 |
| | Lack of time | 91 | 52.6 | 26 | 63.4 |
| | Lack of mentorship | 55 | 31.8 | 17 | 41.5 |
| | Lack of equipment | 51 | 29.5 | 14 | 34.1 |
| | Difficulties in coordinating team | 41 | 23.7 | 11 | 26.8 |
| | Difficulties in obtaining ethical approval | 19 | 11.0 | 8 | 195 |
| | Lack of interest | 18 | 10.4 | 7 | 17.1 |
| | Others | 2 | 1.2 | | |
| Barriers to publishing | Lack of time | 105 | 60.7 | 26 | 63.4 |
| | Lack of financial support | 99 | 57.2 | 27 | 65.9 |
| | Lack of mentorship | 90 | 52.0 | 31 | 75.6 |
| | Limited English proficiency | 64 | 37.0 | 15 | 36.6 |
| | Difficulties in selecting a journal | 39 | 22.5 | 10 | 24.4 |
| | Difficulties in writing the manuscript | 36 | 20.8 | 7 | 17.1 |
| | Difficulties in coordinating coauthors | 30 | 17.3 | 5 | 12.2 |
| | Others | 6 | 3.5 | | |

psychiatry, particularly in leadership positions (Amering et al., 2011; Hart et al., 2019), and Latin America is no exception (Mendlowicz et al., 2011; López-Bassols et al., 2018). Only by recognizing and addressing potential gender-related barriers and opportunities, we can work toward a more inclusive and robust research landscape that better serves the diverse needs of the region.

The present study provides new insights into the experiences, barriers and facilitators to research and academic publications among psychiatrists and psychiatry trainees in Spanish-speaking countries in South America. Nevertheless, the study is not without its limitations. Brazil, a country with a significant and rapidly growing contribution to the global mental health literature (Gonçalves et al., 2009; Moreira-Almeida et al., 2017), was not included. Also, data collection lasted just one month, which very likely prevented a greater sample size from the included countries. Plus, the studied sample does not constitute a statistically representative sample since there was no information from several other countries in the region and data from only psychiatrists and trainees who decided to participate could be included, so the inferential analyses only constitute an exploratory approach to the problem. Likewise, the Nagelkerke $R^2$ values in the regression models suggest that other variables not included in our analyses could have influenced Latin American psychiatrists and psychiatry trainees' participation in research and research dissemination. Further research

should aim to examine this important topic considering other potential variables and with other more robust methodological approaches.

It is important also to note that the present study was focused on psychiatrists and psychiatry trainees. Psychiatrists' main role and contributions to the mental health field have primarily focused on clinical care and service planning and delivery (Bhugra et al., 2015). However, particularly in countries with fewer resources, psychiatrists are increasingly required also to adopt leadership, advocacy and public health roles (Kigozi and Ssebunnya, 2014), roles that are likely also to be reflected in their research interests. Furthermore, the mental health field is complex and various other disciplines contribute to it, from primary care and nursing to anthropology and the social sciences. Future research should also pay attention to psychiatrists' nonclinically focused contributions to the mental health field, as well as to the contributions of nonpsychiatrist researchers working in mental health. At the same time, while our quantitative findings suggest interesting alleys for future research, the present study lacked a qualitative component which could have strengthened our results. Further research on this area could benefit from incorporating a qualitative or mixed-methods approach, which would enable a more in-depth understanding of the experiences of researchers working in the mental health field in the region.

**Table 4.** Logistic regression models for carrying out research as a psychiatrist, publishing as a psychiatrist and publishing as a psychiatry trainee

| Logistic regression model: Carrying out research as a psychiatrist | | | | | | | | |
|---|---|---|---|---|---|---|---|---|
| Variables | B | SE | p | ORa | CI 95% | | $R^2$ | p |
| (Constant) | 0.233 | 0.441 | 0.598 | 1.262 | – | – | 0.106 | 0.007 |
| Research training | −0.387 | 0.300 | 0.197 | 0.679 | 0.377 | 1.223 | | |
| Resource: Time | 0.268 | 0.378 | 0.477 | 1.308 | 0.624 | 2.742 | | |
| Resource: Mentorship | 0.135 | 0.311 | 0.664 | 1.145 | 0.622 | 2.107 | | |
| Barrier: Lack of time | −0.621 | 0.353 | 0.079 | 0.537 | 0.269 | 1.074 | | |
| Barrier: Lack of financial support | 0.732 | 0.295 | 0.013 | 2.080 | 1.167 | 3.706 | | |
| Gender | −0.134 | 0.306 | 0.661 | 0.874 | 0.480 | 1.594 | | |
| Logistic regression model: Publishing as a psychiatrist | | | | | | | | |
| Variables | B | SE | p | ORa | CI 95% | | $R^2$ | p |
| (Constant) | −0.649 | 0.502 | 0.195 | 0.522 | – | – | 0.267 | <0.001 |
| Carrying out research | 1.524 | 0.360 | 0.000 | 4.589 | 2.268 | 9.287 | | |
| Research training | 0.571 | 0.371 | 0.124 | 1.770 | 0.856 | 3.663 | | |
| Barrier: Lack of interest | −0.697 | 0.590 | 0.237 | 0.498 | 0.157 | 1.582 | | |
| Barrier: Difficulties in selecting a journal | 0.572 | 0.482 | 0.235 | 1.772 | 0.689 | 4.556 | | |
| Barrier: Lack of mentorship | −0.143 | 0.377 | 0.704 | 0.867 | 0.414 | 1.813 | | |
| Gender | 0.513 | 0.374 | 0.170 | 1.670 | 0.802 | 3.475 | | |
| Barrier: Lack of financial support | −0.325 | 0.379 | 0.391 | 0.723 | 0.344 | 1.519 | | |
| Logistic regression model: Publishing as a psychiatry trainee | | | | | | | | |
| Variables | B | SE | p | ORa | CI 95% | | $R^2$ | p |
| (Constant) | 2.195 | 1.214 | 0.071 | 8.976 | – | – | 0.371 | 0.004 |
| Barrier: Limited English proficiency | −1.080 | 0.833 | 0.195 | 0.340 | 0.066 | 1.737 | | |
| Barrier: Lack of financial support | −2.581 | 1.151 | 0.025 | 0.076 | 0.008 | 0.722 | | |
| Gender | −1.845 | 1.234 | 0.135 | 0.158 | 0.014 | 1.776 | | |

ORa, OR adjusted; SE, standard error.

## Conclusion

Psychiatrists and psychiatry trainees in Latin America are keen to contribute to the global mental health discussion. Furthermore, they seem to be taking the necessary steps to do so. However, they remain largely underrepresented in the mental health literature. While they continue to face various barriers at a local level, such as lack of funding and protected research time, there also appear to be wider, systematic issues that need to be considered. There is an urgent need to address these barriers so that they can further contribute to the global mental health discussion and, ultimately, help improve the mental health and well-being of the population of Latin America.

**Open peer review.** To view the open peer review materials for this article, please visit http://doi.org/10.1017/gmh.2023.58.

**Data availability statement.** The authors confirm that the data supporting the findings of this study are available within the article.

**Author contribution.** R.R. conceptualized the article. All authors contributed to the survey design, and each one contributed to data collection in their respective countries. J.M.G. conducted the statistical analysis with the help of the SPSS Statistical package. R.R. wrote the first draft of the manuscript. All authors read and approved the final manuscript.

**Financial support.** This research received no specific grant from any funding agency, commercial or not-for-profit sectors.

**Competing interest.** The authors declare none.

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
