## [Reviewer Report]

Dear Professor Gary Belkin, 

I am pleased to submit our manuscript titled ‘Mental health research in Latin America. Barriers, resources, and recommendations’ for consideration in Global Mental Health.

Latin America remains underrepresented in the global mental health discussion. Curious about what factors could be contributing to this issue, we set up this project. We are a group of ten Latin American psychiatrists and psychiatry trainees from nine South American countries that came together for this survey project. As part of the project, we shared a survey with our colleagues from South America. The survey examined barriers to and resources for research and academic publications. Our findings highlight how having access to research training, mentorship opportunities, and protected time for research are significant resources when present. At the same time, they show how lack of funding, protected time for research, research training, and mentorship opportunities are very commonly present barriers in the region, with significant impact. In the manuscript, we discuss some key recommendations and emphasise the importance of ensuring the production, dissemination, and translation of research that will help meet the region’s mental health needs.

No funding was received for this project. The authors have no conflict of interest to disclose. We confirm that this work is original and has not been published elsewhere, nor is it currently under consideration for publication elsewhere. All authors approved the final draft and submission of the present manuscript.

We appreciate your time and consideration.

Sincerely yours,

Rodrigo Ramalho, MD, PhD

---

## [Reviewer Report]

This manuscript presents the results of a quantitative non-probabilistic survey of psychiatrists and psychiatric trainees in Spanish-speaking Latin American countries regarding barriers and resources for research. The survey was completed by 214 participants and identified many involved in research and publications, and most acknowledged problems including lack of funding, time, and mentorship.

I believe that the topic is of extreme relevance in global mental health, as representation and contributions from Spanish-speaking Latin America are very scarce in this research area, while the mental health needs in this part of the world are substantial. This manuscript presents a preliminary approach to investigating and addressing the issue by identifying key barriers and resources in a non-systematic approach. I believe that this manuscript is relevant and contains valuable research that is presented acknowledging its important limitations.

I have some minor comments and suggestions:

As for the title, I wonder to what extent it could be made more specific while maintaining its broader appeal and not making it too long. While the title refers to mental health research in general, this research is limited to research by psychiatrists and psychiatric trainees, which are essential but not unique actors in mental health research. In addition, the study is limited to Spanish speaking countries, which are of course majority but not the totality of Latin America. In addition, ‘recommendations’ is added to the title, although it was not part of the research survey itself - rather, recommendations by the authors are given at the discussion based on the results, as usual in research papers. My recommendation would be to make the title more specific to research by psychiatrists (and psychiatric trainees) in Spanish-speaking Latin America, and removing ‘recommendations’* from the title.

*If authors would like to give more relevance to the ‘recommendations’ section and add it to the title, details should be given in Methods as to how those recommendations were elaborated, and that part of the study should have corresponding mentions in the Introduction and Results; I understand that would be a major change, and time consuming.

It would be good to add to the Discussion some lines related to two aspects that are both limitations, recommendations for future research, and considerations. On the one hand, the fact that mental health research includes contributions from actors other than psychiatrists and psychiatric trainees (including other physicians such as primary care, public health physicians, neurologists, and other specialists and non-specialists; and more abundantly from non physician actors) - which is a limitation of this survey buyt also an opportunity for authors in this manuscript to discuss briefly why mental health research by psychiatrists and trainees is important and what is its specific role. On the other hand, the quantitative results provided by authors indicate some general themes but is not able to go deeper, for which qualitative and mixed-method studies are needed - this is both a limitation but also an opportunity for authors to discuss the way to go for future research.

Finally, I must admit that I was not able to find the manuscript tables anywhere in the journal review portal. I do not know where the problem lies but would like to take a look at those while re-reviewing the paper.

If convenient, I could also send authors through the journal editorial managers an annotated .pdf version with some language changes and suggestions - I do not see in this reviewing portal a way to add reviewing attachment files. There are some minor English language issues worth improving.

Thank you for the opportunity to review this manuscript.

---

## [Reviewer Report]

This is a pertinent study on an area in need of deeper explorations,and concrete improvements for Latin American psychiatry. Well conceived, comprehensive and informative, the text encompasses a number of features that are in need of substantial changes in order to improve the image of MH and psychiatric research in the sub-continent, and its visibility around the world.

This reviewer’s main observations and comments follow:

1) As Brazil is not included in the study and, yet, is probably the Latin American country with major psychiatric research, it could be convenient to include some useful information from suthorized sources (Jair Mari, Rubim de Pinho and other authors) and their impact on the whole Latin American picture.

2) Another limitation, in addition to lack of representativeness of the sample (acknowledged by the authors) is th fact that the survey lasted only one month, thus possibily depriving the authors of a greater size and more productive search of data.

3) In the Results section, it would be pertinent to comment on two interesting additional findings: the higher number of women in the sample, and the fact that only 3 of the 9 participating countries have 3/4 of the total sample; furthermore, the sample shows most psychiatrists from Peru, Argentina and Uruguay, even though it is well know that Colombia and Chila also have good number of specialists.

4) The point of higher humber of women is repeated in Resuts Section, p.4, par. 1, l. 2 and 5. The same happens with the highest number of participants coming from public universities (Sectin Results, par 1, l. 6 and par 2, l. 1. It is important to avoid text and word duplications.

5) In p. 6, par 2, l. 6 the sentence “difficullties coordinating a research time was an advantage” (for conducting research) sounds illogical, so it will have to be better explained or modified.

6) Tha last paragraph of the Discussion section could be broadened a little bit and go under the subheading of Conclusions. Similarly, the alphabetical order of the Bibliography will have to be corrected as an author starting with “A” comes after another whose name starts with a “C”.

---

## [Reviewer Report]

The study is focused on a relevant theme and the methods used were adequate. As recognised by the authors, the sample is not representative: some of the more productive countries in mental health-related research (e.g. Brazil and Mexico) are not included and there is a predominance of the psychiatrists with less years of working experience in psychiatry. Despite this important limitation, this is an important exploratory paper, as it gives a preliminary insight of the resources and barriers for research in mental health in countries with less resources in Latin America.

I believe that a few changes might improve the paper.

The paragraph in page 4, lines 40-49 is difficult to read. I suggest to rewrite this paragraph in a simpler and more coherent way.

I think the paragraph in Page 7, lines 29 to 52, on the contribution of papers from Latin American countries to global mental health-related literature, should be moved to the Introduction section.

I suggest to include some specific recommendations on what could specifically be done to overcome the barriers found by Psychiatry trainees.

---

## [Reviewer Report]

Dear Authors:

Please, check the reviewers' recommendations and suggestions following each one or explaining why you do not agree with.

Also, please review the manuscript with a software for similarity (e.g. Turnitin) for checking the writing.

Thank you.

---

## [Reviewer Report]

Dear Dr Judith Bass and Prof Andrés Fandiño-Losada,

Thank you for the opportunity to resubmit our manuscript – previously – titled ‘Mental health research in Latin America. Barriers, resources, and recommendations’ (GMH-23-0099) for consideration in Global Mental Health.

We are also grateful to the reviewers for their constructive feedback and suggestions, which have undoubtedly improved the quality of the manuscript.

Sincerely yours,

Rodrigo Ramalho

---

## [Reviewer Report]

Thank you for revising your manuscript - I believe the revisions are appropriate and address the reviewers' comments (certainly mine).

I would like authors to add a mention to the logistic regression analysis at the end of Methods section, as it is lacking. I would also suggest authors to mention briefly in the discussion whether it may be worthy to incorporate stratified analysis by sex in future research - it seems that in general there are disparities in terms of academic leadership and production, and that those may be more accentuated in LMIC due to financial and/or cultural factors, and it would be interesting to explore this in Latin America.

Those are minor issues, and if the editor considers that authors should address them in another revision, I think that afterward the editor may consider making a decision without further peer review.

---

## [Reviewer Report]

The changes that were made in the fist version significantly improved the paper. It is now an interesting paper that reads very well.

I just recommend to correct two minor errors:

On page 5, lines 19 - 20 ("...having 1-5 years of working experience as psychiatrists (versus n=41; n=23.7% with >25 years....") the n= before 23.7% should be deleted

On page 5, lines 47- 49 ("Most psychiatry trainees reported were studying at a public university (n=33; 80.5% versus n=8; 19.5 in private universities"), 19.5 should be followed by %.

---

## [Reviewer Report]

Dear authors:

The reviewers have suggested methodological revisions which you should follow.

In this manner, the explanations on the logistic regressions should be improved in the methods sections, gender and Psychiatrists vs. Trainees status should be included as explanatory variables. 

You should present the results of corrected logistic regressions in the results sections using the appropriate tables and descriptive texts and discuss these results in the discussion section.

Please, discuss the selection biases related with who have answered the survey and also discuss the response proportion.

Thank you.

---

## [Reviewer Report]

Dear Dr. Judith Bass and Dr. Andrés Fandiño-Losada, 

We are thankful for the opportunity to resubmit our manuscript ‘Mental health research in South America. Psychiatrists and psychiatry trainees’ perceived resources and barriers’ (GMH-23-0099.R1) for consideration in Cambridge Prisms: Global Mental Health.

We would also like to thank again the reviewers for their time and consideration. Their comments and feedback have significantly improved the quality of the manuscript.

Kind regards, 

Rodrigo Ramalho